# Influence of Fiber Addition on the Properties of High-Performance Concrete

**DOI:** 10.3390/ma14133736

**Published:** 2021-07-03

**Authors:** Szymon Grzesiak, Matthias Pahn, Milan Schultz-Cornelius, Stefan Harenberg, Christoph Hahn

**Affiliations:** 1Department of Civil Engineering, Technical University of Kaiserslautern, 67663 Kaiserslautern, Germany; szymon.grzesiak@bauing.uni-kl.de (S.G.); milan.schultz-cornelius@bauing.uni-kl.de (M.S.-C.); 2Implenia Schalungsbau GmbH, 67240 Bobenheim-Roxheim, Germany; stefan.harenberg@implenia.com; 3Master Builders Solutions Deutschland GmbH, 68199 Mannheim, Germany; christoph.hahn@mbcc-group.com

**Keywords:** high performance fiber reinforced concrete (HPFRC), polypropylene fiber (PP), polyvinyl alcohol fiber (PVA), compressive strength, residual flexural strength, splitting tensile strength

## Abstract

High performance fiber-reinforced concrete (HPFRC) has been frequently investigated in recent years. Plenty of studies have focused on different materials and types of fibers in combination with the concrete matrix. Experimental tests show that fiber dosage improves the energy absorption capacity of concrete and enhances the robustness of concrete elements. Fiber reinforced concrete has also been illustrated to be a material for developing infrastructure sustainability in RC elements like façade plates, columns, beams, or walls. Due to increasing costs of the produced fiber reinforced concrete and to ensure the serviceability limit state of construction elements, there is a demand to analyze the necessary fiber dosage in the concrete composition. It is expected that the surface and length of used fiber in combination with their dosage influence the structure of fresh and hardened concrete. This work presents an investigation of the mechanical parameters of HPFRC with different polymer fiber dosage. Tests were carried out on a mixture with polypropylene and polyvinyl alcohol fiber with dosages of 15, 25, and 35 kg/m^3^ as well as with control concrete without fiber. Differences were observed in the compressive strength and in the modulus of elasticity as well as in the flexural and splitting tensile strength. The flexural tensile strength test was conducted on two different element shapes: square panel and beam samples. These mechanical properties could lead to recommendations for designers of façade elements made of HPFRC.

## 1. Introduction

Since the development of concrete, RC constructions allow for more and more filigree and lightweight elements with the contemporary growth of structure loads [1]. Reduced cross-sections of components are associated with advanced technologies and materials based on higher material properties [2]. With an increase in concrete specifications like compressive strength, the post crack behavior of concrete becomes worse. In concrete compositions, different kinds of fibers are added to avoid brittle fracture behavior and ameliorate the ductility of those materials [3]. Fiber reinforcement concrete (FRC) has already been used successfully in many horizontal and vertical structural as well as non-structural elements [4]. For example, using fiber reinforcement together with traditional steel bar reinforcement decreases crack propagation and displacement of concrete slabs like industrial floors [5]. In buildings and bridges in seismic areas, fiber reinforced concrete improves the behavior of structural parts like columns, beams, or walls [6].

Recently, FRC has also been used for the production of pre-cast elements in which the fibers—in combination with ultra-high-performance concrete—enhance the durability of cracked concrete [7]. As studies show, a decisive role is played by the dense microstructure of concrete pre-cast elements. This can be ensured by low water–cement and water–binder ratios. Mateckova et al. [8] report that high-performance concrete with its dense structure presents higher resistance to chemical penetration in comparison to ordinary concrete. The character of used materials in HPC can improve the acid attack by FRC through better integrity of the binder matrix to fiber inclusion. Ali et al. [9] showed the influence of fibers and silica fume on the mechanical and durability performance of concrete concerning a reduction in the materials’ permeability.

The possible benefits of using HPFRC are in sustainable resource management. A good example for the use of FRC are façade panels in building constructions, leading to a considerable reduction in the material volume [10]. Therefore, the major advantage of fiber-reinforced concrete elements is the reduced thickness, thus leading to a reduction in CO_2_ footprint. Without steel rebars, façade panels can be just a few centimeters thick [11]. For concrete elements without steel reinforcement, a corrosion protection system like concrete cover can be omitted [12].

Vertical exterior elements of buildings exposed to environmental factors are investigated for structural performance under gravity and wind load. Concrete elements such as façade panels for certain boundary conditions are under flexural load. The wind pressure is distributed as area load, which causes tensile and compressive stresses in the cross-section of the building’s façade and results in deformation [13]. Therefore, exterior elements are designed to transfer loads to the main structural system of the building. This is why the flexural tensile strength of fiber reinforced concrete is one important design parameter [14]. Moreover, the impact of panel behavior and damage to the FRC is evaluated as a safety factor.

To understand the behavior of HPC and UHPC as well as concrete in general with fiber addition, scientific measurement methods in experimental research study have recently been published [15]. Typical fibers for HPFRC are made of steel [16], carbon [17], or polymers [18]. Examining the available research literature demonstrates that another material like wool [19], basalt [20], or glass [21] could be successfully added to concrete. The fibers differ according to their origin, mechanical properties, and their corrosion resistance [22]. The analysis of this material focuses particularly on a unique combination of concrete and fiber reinforcement [23]. The damage process and the mechanical properties of HPFRC can be taken into account for different dimensions and different shapes of samples [24]. Results of the experimental tests should implement the anisotropy of fiber orientation in the concrete matrix [25]. The location of deformation and the position of the cracked zone can lead to difficulties during examination [26]. By means of a clip gauge, it is possible to estimate the behavior of specimens in the cracked region. The clip gauge is used to measure the crack mouth opening displacement (CMOD) [27]. The details of the classic experimental setup and the examination with a clip gauge will be presented in the Section 2.

The aim of this work was to investigate the influence of fiber addition on the properties of high-performance concrete. As is known from other publications, the addition of fiber does not always have a positive effect on the mechanical properties. Other studies have also focused on other types of fibers in connection with HPFRC. Therefore, in this study, it was proposed to also consider the fiber type and shape variation of the concrete sample. Furthermore, this paper expands the database with an overview of the mechanical properties in HPFRC with polypropylene and polyvinyl alcohol fiber. It is suspected that the surface and length of used fiber influence the structure of fresh and hardened concrete. Optimization and a better quantity of fiber dosage will allow for a reduction and also a better use of the materials in the concrete mix design. Mechanical parameters of HPFRC enable the economical design of filigree, safe, and lightweight elements.

## 2. Experimental Program

### 2.1. Materials

In this study, only fine aggregates and particles were used to improve the homogeneity of high-performance concrete [28]. In the following sections, the properties of the fibers and the concrete mix design are discussed.

#### 2.1.1. Fiber

In this study, the influence of two fiber types on the fresh concrete and the mechanical properties of hardened concrete is investigated: polypropylene fibers (PP) and polyvinyl alcohol fibers (PVA) (see Figure 1). The characteristic of PP- and PVA-fibers are given in Table 1. Compared to steel, both fibers are corrosion-resistant.

#### 2.1.2. Concrete Mix Design

For this experimental research, five different concrete mixtures with the same amount of cement, aggregates, and additives (Table 2) were prepared. They only differed in the fiber type (Table 1) and fiber dosage (Table 3). Mix ID 1 contained 35 kg/m^3^ of the MasterFiber 401 (PVA). For comparison with the MasterFiber 235 SPA (PP), Mix ID 2 contained 35 kg/m^3^. To investigate the influence of the fiber dosage, Mix IDs 3, 4, and 5 contained 25 kg/m^3^, 15 kg/m^3^, and 0 kg/m^3^ of fibers, respectively. The mixtures were prepared by using a 55-L-capacity horizontal forcing type concrete mixer.

The appropriated HPFRC was produced with cement CEM I 42.5 R [29], quartz sand 0.1/0.6 [30], limestone powder, and silica fume. The limestone powder and the silica fume provide a dense microstructure of concrete and are used as fillers. The aggregates were dried sand and basalt [31]. Figure 2 presents the grain size curve of the material used in the present study [32]. For better workability, a plasticizer, MasterGlenium ACE 430, was used [33]. The water–cement (w/c) and water–binder (w/b) ratio in the concrete composition was 0.323 and 0.238, respectively.

### 2.2. Testing Procedure

In this study, tests were carried out on fresh and hardened high-performance concrete with and without fibers. The compressive strength tests on the hardened concrete were performed on 150 mm cubes according to EN 12390-1 [36]. Cylinders of 150 mm in diameter and 300 mm in height were used to test the modulus of elasticity and splitting tensile strength according to EN 12390-3 [37].

A test program based on the different fiber dosage was conducted (Table 3). The flexural tensile strength test was conducted on square panel specimens (250 × 250 × 35 mm^3^) in accordance with EN 12467 [38] and beam specimens with a cross-section (100 × 100 × 400 mm^3^) in accordance with EN 14651 [39]. For comparison purposes in both tests, we used the classical measurement method with the force–displacement relationship. Simultaneously, for the measurements of beam deformation, we applied the test method of crack mouth opening displacement (CMOD), which has been recently mentioned in the experimental investigations of HPFRC [27,40]. For this purpose, the bottom surface of the beam specimens had a notch with a depth of 17 mm and a width of 5 mm according to the procedure in EN 14651 [39] (see Figure 3). The notch was milled for the determination of strain during the crack initiation in the middle of the span. Near the notch were glued metal plates to attach the clip gauge (see Figure 4). The clip gauge measures the displacement between two points that are on two different edges of the crack [41].

As shown in a study by Bi et al. [42], the distribution of fiber during concrete flow influences the mechanical properties of HPFRC. Mostly, fibers are randomly distributed but computer tomography shows that fibers accumulate in the upper part of the specimens. It has been observed that the flexural strength of HPFRC is higher with more fibers located in the tension zone. Therefore, during the bending tests, the upper side of the beams is turned backwards.

Three-point bending tests were carried out on plates and beams in order to be able to estimate the flexural strength [38]. The stresses $f_{L}$ were calculated according to the *fib* Model Code 2010 [43], as follows:(1)$$f_{L}=\frac{3\cdot F\cdot l}{2\cdot b\cdot h^{2}}$$
where F represents the cylinder force in [N]; b and h are the width and height of the specimens, respectively; and l is the distance of the supports. The distance for the plates equaled l=0.75·L and for the beams, l=b/0.3. Dimensions are provided in [mm].

In their study, Schultz-Cornelius [24] investigated flexural strength for many different thicknesses of façade panels separately. They observed an effect in size for specimens with a depth below 50 mm. The tests showed a linearly increasing bending tensile strength with decreasing thickness. In order to examine the impact of fiber dosage on the bending tensile strength, only panels with a thickness of 35 mm were tested.

The specimens were cured and stored at a temperature of 20 °C and a relative air humidity of 60%. Each test on hardened concrete was performed after 30/31 days. The measurements were processed by Catman AP software. All bending tests were performed based on controlled displacement with the same testing speed and the same measuring rate of clip gauge.

## 3. Analysis of the Results

In this study, the properties of fresh concrete as well as the mechanical parameters of hardened concrete like compressive strength, modulus of elasticity, bending, and splitting tensile strength were investigated. The test results are discussed in the following sections.

### 3.1. Properties of Fresh Concrete

The slump test was performed according to EN 12350-5 [44]. The results presented in Table 4 show that both PP and PVA fibers performed well during the slump flow test. The slump flow diameter was measured on the base plate in two directions, from which the average value was calculated. It should be mentioned that the plasticizer MasterGlenium ACE 430 dosage in each concrete mixture was identical. With increasing fiber dosage, the workability of HPFRC increases. Tests show that the slump flow measure increased from 595 mm to 650 mm based on a PP fiber dosage of 15 kg/m^3^ and 35 kg/m^3^. The phenomenon of higher fiber dosage and a concurrently larger slump flow diameter was also observed in investigations of self-compacting concrete with PP and steel fiber batches [45]. For HPFRC with PVA fibers, a smaller slump flow diameter is recognized than for HPFRC with PP fibers. If a PVA fiber with a smaller diameter is used, the mixtures tend to absorb much more water and hence change the consistency of fresh concrete. This is due to the high specific surface area of PVA fibers. Chen [46] reported that fine fibers were responsible for reducing the mixture workability and suggested a combination of small and medium fibers for the best balance of fresh concrete and hardened material properties. The progress of the slump experiment and a picture of a fresh concrete mixture are shown in Figure 5.

Table 4 shows the air depending on the fiber dosage of each mixture. This was determined according to EN 12350-7 [47]. The concrete with less pore volume had a higher bulk density. Furthermore, the air impacts mechanical properties like compressive strength and modulus of elasticity. Hassan [48] stated that concrete with lower porosity was resistant against chloride penetration. Furthermore, it showed a lower fluid and gas permeability, which impacts the frost resistance of HPFRC.

### 3.2. Compressive Strength

Figure 6 shows the compressive strength of the concrete after 28 days with different fiber dosages. The results showed that the compressive strength decreased with increasing fiber dosage. Lower dosages of fiber in normal concrete, in contrast, behave differently (e.g., under 30 MPa). In general, the mechanical properties of concrete with a lower compressive strength will be improved through the addition of fibers [49,50]. The space between concrete and fibers is filled with air, which already starts to accumulate during the mixing process between the adhesive surfaces. The air voids are linked to the small interaction between the PP and the cement paste. A small air volume reduces the compressive strength of high-performance concrete. Furthermore, the higher fiber dosage reduces the volume of the concrete–matrix dosage and at the same time reduces the compressive strength and modulus of elasticity of the concrete.

### 3.3. Modulus of Elasticity

The modulus of elasticity was determined according to EN 12390-13 [51]. Figure 7 shows the test results for the cylindrical specimens [36]. The modulus of elasticity’s value depends on the fiber dosage. An increase in the volume fraction of PP fiber in concrete leads to a decrease of the modulus of elasticity, which is due to the compressive strength of HPFRC. Similar results were obtained in a study on fiber reinforced concrete with a compressive strength up to 100 MPa [52].

### 3.4. Splitting Tensile Strength

The addition of fibers to the concrete mix has a major impact on the tensile strength of FRC. The test results showed an increase from 4 MPa for concrete without any fibers to 6.9 MPa for concrete with a fiber dosage of 35 kg/m^3^ (see Figure 8). Furthermore, the results showed that the difference in the splitting tensile strength between a fiber dosage of 25 kg/m^3^ and 35 kg/m^3^ was lower compared to a fiber dosage of 15 kg/m^3^ and 25 kg/m^3^. It can be assumed that this effect is caused by the strength of the fiber–matrix interface. The strength depends on the volume of the concrete surrounding the fibers. With increasing fiber dosage, this volume decreases and hence the pull-out-strength of the fibers in the cracking zone is reduced. It can therefore be concluded that the splitting tensile strength was at most 72.5% higher due to the addition of fibers.

### 3.5. Residual Flexural Strength

For façade panels, the most important mechanical property is the bending tensile strength [53]. Fibers in concrete enhance the concrete’s mechanical properties. They absorb post-crack energy and improve ductility of the FRC. For thin concrete elements like façade panels, a higher bending tensile strength leads to an improved resistance against area loads caused by wind and impacts.

The experimental setup and the dimensions of the test specimens are discussed in Section 3. The plates according to EN 12467 [38] reached higher bending-tensile strengths up to 10.5 MPa for a fiber dosage of 35 kg/m^3^. The test results with average values for plates with PP fiber are presented in Figure 9.

The stress–deflection curves in Figure 9 show a linear behavior at the beginning of the loading. After the first crack, stress deflection curves are non-linear. In HPFRC, the deflection, and concurrently the stress, still increases. Once the maximum value of stress has been reached, they decrease slowly. A similar effect of decreasing stresses after the first crack was described by the Association Francaise de Genie Civil [54] (AFGC). In concrete without fibers, the stress–deflection curves reached 5.61 MPa after the first crack and then collapsed. In contrast, the specimens with fiber dosage had an inclined plateau phase. This behavior is facilitated by a bond length of the PP fibers in the fiber–matrix interface, which is activated as soon as a crack occurs. The longer the bond of the anchored fiber, the greater the pull-out force. This behavior is the so-called bridging effect [55]. According to the first crack [38], it was possible to estimate the bending tensile strength of the concrete with fiber and concrete without fiber. The results with different fiber dosages are presented in Table 5.

The test results in Table 5 clearly reflect the addition of fiber dosage: an increasing bending tensile strength was observed with higher fiber dosage. The tests revealed an increase of up to 4% for a fiber dosage of 15 kg/m^3^, 15% for 2d5 kg/m^3^, and 18% for 35 kg/m^3^. The experiments showed that by adding fibers, the failure mode changed from brittle to ductile. Similar to the study conducted by Kahanji [56], the variation of the fiber dosage has an enormous influence on the post-crack behavior of HPFRC.

The same results as for the panels were achieved with beams according to EN 14651 [39]. Figure 10 presents the test results with CMOD and Figure 11 shows the results with a deflection in the middle span of the specimen. The stress–strain diagram shows that the strain increased immediately after crack initiation. A comparison of the test results obtained for mixtures ID 2 (35 kg/m^3^) and ID 3 (25 kg/m^3^) showed similar bending tensile strengths. However, the dosage of fiber quantity for mixtures ID 2 and 3 differed. By comparing these two fiber dosages, an increase in the bending tensile strength of up to 200% was evident. Mixture ID 4 showed increases of up to 84%. The strain of mixture ID 4 measured with the clip gauge reached a plateau. Following this crack initiation, the measured strains increased with higher load capacity. For concrete without fibers (mixture ID 5), the stress–strain curve reached 3.14 MPa, then declined and reached zero.

The test method for fiber reinforced concrete presented in EN 14651 [39] enables the calculation of flexural tensile strength. The stresses $f_{L}$ in limit of proportionality (LOP) were calculated according to the equation for the test method in concrete with metallic fiber [39,57]. The required force was determined in the case of crack opening CMOD = 0.05 mm (see Figure 12). The flexural tensile strength (limit of proportionality) was calculated and listed in Table 6 (characteristic values with k_s_ = 2.336). The correlation of the stress–strain curves showed that with a fiber addition between 15 kg/m^3^ and 35 kg/m^3^, the results of LOP were similar. A significant difference was found in the post crack behavior due to the addition of fibers to the mix. The samples with Mix ID 5 failed when the maximum flexural stresses were reached. This means that the CMOD with a value higher than 0.5 mm could not be determined.

Based on experimental stress–strain curves, parameter $f_{L}$ was evaluated at four different CMOD values: 0.5, 1.5, 2.5, and 3.5 mm. The residual bending tensile strength $f_{L}$ for different fiber dosages was calculated and listed in Table 6 (characteristic values). This study allows for the following conclusions to be drawn in post-crack bending tensile strengths. With a fiber addition of 25 kg/m^3^ and 35 kg/m^3^, the maximum bending tensile strength varied between 6.30 MPa and 5.66 MPa for CMOD of 0.5 mm and 9.24 MPa–7.34 MPa for CMOD of 3.5 mm. For the fiber addition of 15 kg/m^3^, the residual post-cracking strength reached 3.06 MPa for CMOD of 0.5 mm and 4.94 MPa for CMOD of 3.5 mm.

Figure 13 shows the stress–deflection curves of different fiber types. The first curve reflects the long MasterFiber 235 SPA (PP), and the second the short MasterFiber 401 (PVA). The short fiber is also thinner than the long fiber, which affects the mix design with a much higher number of fibers in the concrete. Both concrete mixtures were made of the same raw materials and comprised a fiber dosage of 35 kg/m^3^. Although the PP fiber was longer than the PVA fiber, a similar maximum bending tensile strength was achieved with both types of material, regardless of the fiber length. In the study by Yoo [58], the authors reported that the use of a longer fiber led to higher flexural strength than the shorter fibers. In the experimental test, only strains as a function of the length of the fiber were detected. The stress–deflection curves for MasterFiber 235 SPA (PP) showed significantly better results than the stress–deflection curves for MasterFiber 401 (PVA) following a deflection of 6 mm. The stress–deflection curve of fiber type MasterFiber 401 (PVA) revealed a higher increase in deflection compared to fiber type MasterFiber 235 SPA (PP), which may indicate unfavorable adhesion forces between PVA fiber and the matrix. Shorter fibers pull-out of the matrix faster than longer fibers. This is attributed to the bonding forces between the fibers and the concrete matrix. The 30 mm long fibers provided a better friction range than the 12 mm long fibers and also provided a better stress transfer in the matrix. However, the smaller fiber had a higher tensile strength. Different types of fibers reflect the bonding behavior between the fibers and the surrounding concrete [59].

Figure 14, Figure 15 and Figure 16 clearly show the cracking patterns of selected specimens in a three-point bending test. The main crack occurred in the middle of the specimen. During the loading tests, the crack width increased and lead to breakage or pull-out of the fiber. Due to the high strain rate of the materials, two halves of a concrete slab held together. There was no brittle failure because PP or PVA fibers under bending tensile load prevented the opening of a crack and finally prevented the sudden destruction of the concrete. The fibers, which were distributed along the axis of the beam, improved the bending tensile strength of the concrete. Some of the fibers in the crack zone reached their tensile strength.

## 4. Discussion

As tests of the fresh and hardened concrete show, HPFRC is strongly affected by the density of used fiber and the presence of the air voids. The bulk density, compressive strength, and modulus of elasticity decreased with fiber addition. This effect can be attributable to fiber dosage in the concrete mix. Liu et al. [60] analyzed the permeability of carbon fiber reinforced concrete and observed the same impact based on the water–cement ratio. The fiber improves the impermeability of concrete only for the w/c ratio of 0.25. The higher ratio reduces the impermeability of hardened concrete. Richardson [61] studied the differences between plain concrete and concrete with fiber additions. Due to the higher air content in the fiber-reinforced mixtures compared to normal concrete, the compressive strengths differed from each other.

Another tendency showed in some cases of the normal strength concretes where the higher dosage increased compressive strength and modulus of elasticity. Kilmartin-Lynch et al. [62] tested a concrete mix with a compressive strength of 50.34 MPa and reached higher values for the dosage of recycled polypropylene fibers. The increase of higher fiber dosage can be noted because the fibers became more densely spaced, which therefore increased compressive strength. Sekhar Das et al. [63] showed that the compressive strength of 36.9 MPa initially increased with fiber content up to 0.5% and then decreased with further use of fibers. This content corresponds to a fiber dosage of 4.55 kg/m^3^, which was not tested in the presented study. Therefore, the conclusion is that the correlation between compressive strength and modulus of elasticity depends on fiber dosage. For further studies, a lower fiber content should be tested.

In the experiment, the fiber dosage improved the flexural properties of concrete. The flexural strength increased the maximal 31% for a fiber dosage of 25 kg/m^3^ in comparison to the plain concrete. Rostami et al. [64] reported that the highest flexural strength was 94% relative to the control sample. Such a large difference may be due to the length of the polypropylene fiber. In their experiment, Rostami et al. used longer PP fibers of 48 mm. Similar observations were made by Zhou et al. [65] who reported that PP improved the flexural strength of concrete. Unfortunately, the tests were carried out on another type of fiber with a maximal length of 18 mm.

Not only are the values of the flexural strength reflected in the studies on FRC, but the failure modes for specimens in flexural tests show similarities with other research. Abbas et al. [66] tested tunnel lining segments using UHPC with fiber dosage where the crack developed during the progression of load in the middle part of the sample.

## 5. Conclusions

The objective of this work was to investigate how fiber dosage affects the mechanical parameters of high-performance fiber reinforced concrete. This study allows for the following conclusions to be drawn in the area of material properties:The percentage of air voids in the concrete corresponds to the compressive strength and the modulus of elasticity of the concrete. A significant difference was found in the compressive strength of the concrete due to the addition of fibers to the mix. The fiber addition of 15 kg/m^3^ in the concrete composition reduced the compressive strength from 83.2 MPa to 79.6 MPa. The higher fiber dosage showed a similar trend. Furthermore, it reduced compressive strength and the modulus of elasticity of the concrete.PP and PVA fibers have proven to be effective in increasing the splitting tensile strength of concrete, which allows better utilization of material capacities and has an impact on the production costs of FRC members. The comparison showed that the dosage of fibers increased from 4.0 MPa to 5.0 MPa (for 15 kg/m^3^), 6.7 MPa (25 kg/m^3^), and 6.9 MPa (35 kg/m^3^).The analysis of the bending tensile tests revealed differences between MasterFiber 401 (PVA) and MasterFiber 235 SPA (PP) in the post-crack phase. MasterFiber 235 SPA is intended to be used because of its higher ductility.The bridging effect, which improves the safety of the concrete components, was identified in the bending tensile test. The bending tensile strength of concrete with added fibers increased by up to 18% compared to materials without fibers. Some of the fibers reached their tensile strength and were no longer involved in the transfer of the load. The pull-out effect of the fiber changed the brittle fracture behavior of concrete into the ductility behavior of these materials.In the present study, the highest PP fiber dosage examined in the concrete composition amounted to 35 kg/m^3^. However, the addition of more than 25 kg/m^3^ of fibers to the concrete mix had less influence on the bending tensile strength of the concrete. This concrete mix had an overcritical fiber dosage and was characterized by tensile strain-hardening behavior. A comparison of the stress–deflection curves with the addition of 25 kg/m^3^ and 35 kg/m^3^ of fibers revealed that the cracking behavior of concrete for these two fiber contents did not differ significantly.Further study of HPFRC comprises more mechanical experiments. New attempts will be focused on anchorage techniques for façade plates in building construction. A higher load capacity for the steel anchor system with a higher fiber dosage is expected.

## Figures and Tables

**Figure 1 materials-14-03736-f001:**
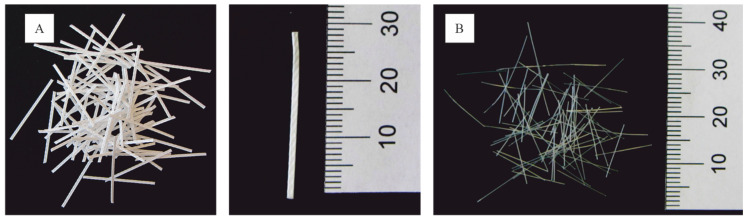
(**A**) Polypropylene fiber MasterFiber 235 SPA (PP). (**B**) Polyvinyl alcohol fiber MasterFiber 401(PVA) used in the present study.

**Figure 2 materials-14-03736-f002:**
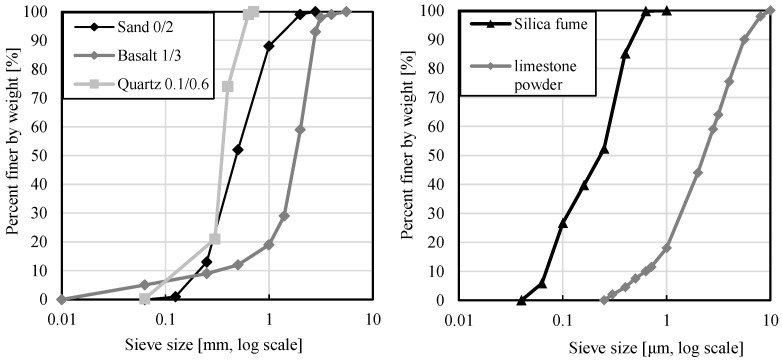
Grading of the fines for the material used in the present study. Sand, basalt, and quartz own studies. Silica fume [34], limestone powder [35].

**Figure 3 materials-14-03736-f003:**
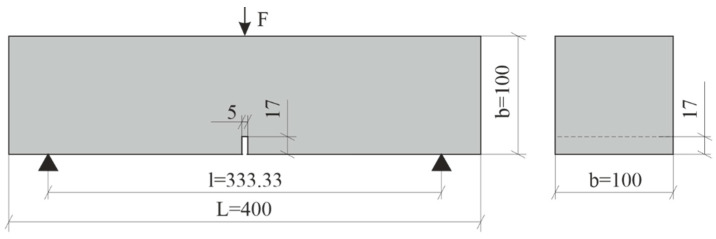
Beam specimen geometry. Dimensions in [mm].

**Figure 4 materials-14-03736-f004:**
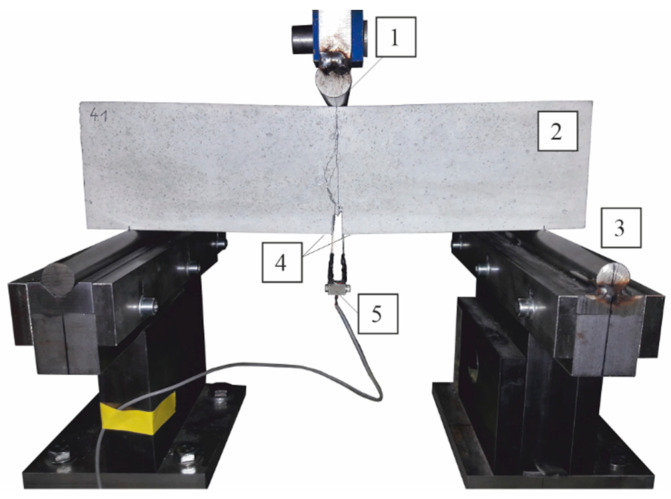
Experimental setup: 1 = point of application of symmetrical load, 2 = HPFRC beam specimen, 3 = supports, 4 = glued on metal plates to attach the clip gauge, 5 = clip gauge.

**Figure 5 materials-14-03736-f005:**
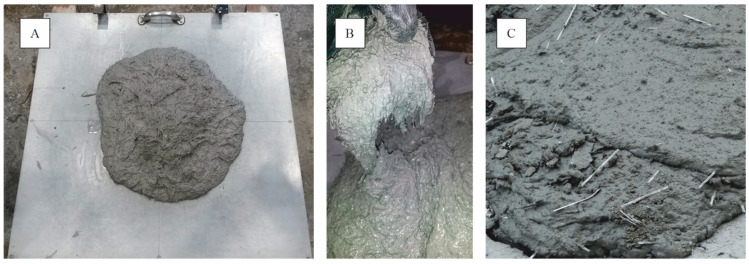
(**A**) Slump experiment for MasterFiber 401 (PVA). (**B**) Fresh concrete with MasterFiber 401 (PVA). (**C**) Fresh concrete with MasterFiber 235 SPA (PP).

**Figure 6 materials-14-03736-f006:**
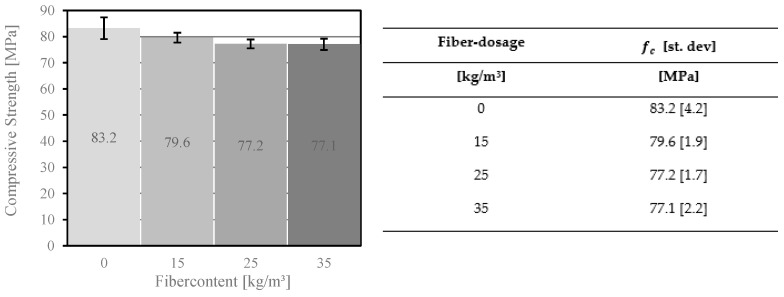
Compressive strength results of concrete with different fiber dosages of MasterFiber 235 SPA (PP).

**Figure 7 materials-14-03736-f007:**
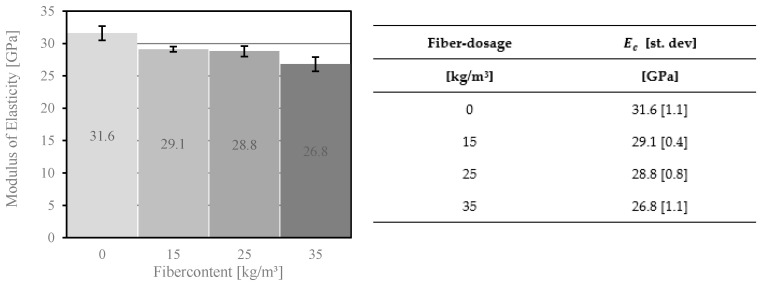
Modulus of elasticity of concrete with different fiber dosages of MasterFiber 235 SPA (PP).

**Figure 8 materials-14-03736-f008:**
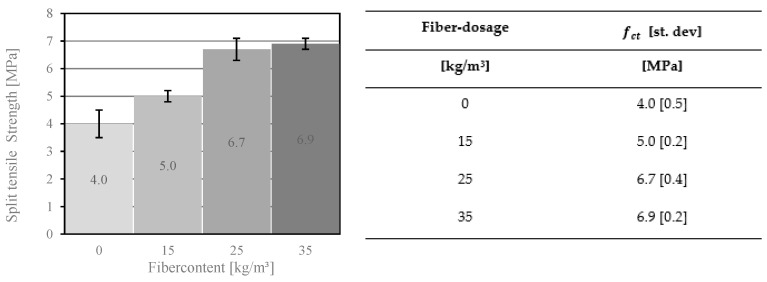
Split tensile strength of concrete with different fiber dosages of MasterFiber 235 SPA (PP).

**Figure 9 materials-14-03736-f009:**
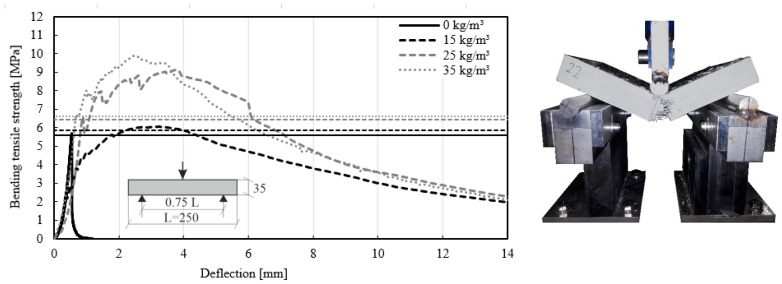
Bending-tensile strengths of concrete with different fiber dosages MasterFiber 235 SPA (PP) in accordance with EN 12467 [38].

**Figure 10 materials-14-03736-f010:**
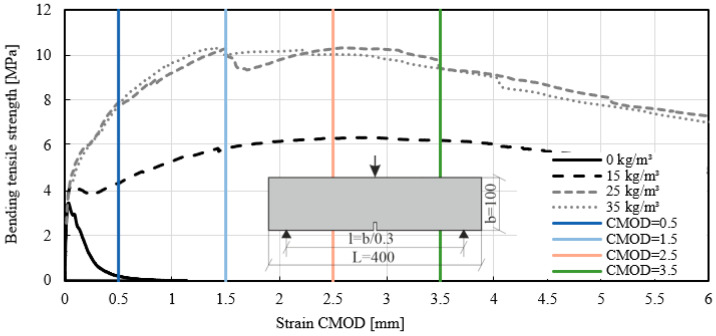
Stress–strain curves of the beam test-series with different fiber dosages MasterFiber 235 SPA (PP). Clip gauge strain in accordance with EN 14651 [39]. The vertical axes are CMOD 0.5 to CMOD 3.5.

**Figure 11 materials-14-03736-f011:**
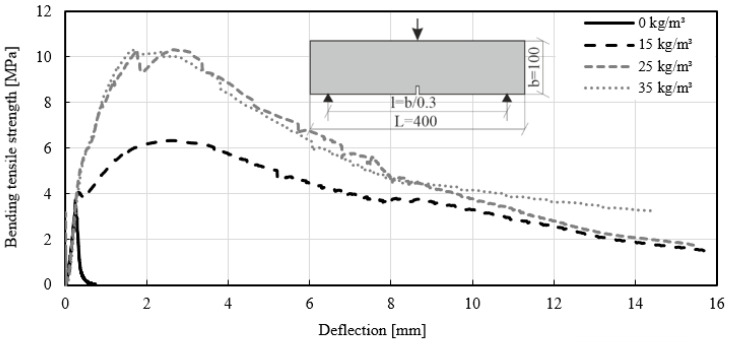
Stress–deflection curves of the beam test-series with different fiber dosages MasterFiber 235 SPA (PP).

**Figure 12 materials-14-03736-f012:**
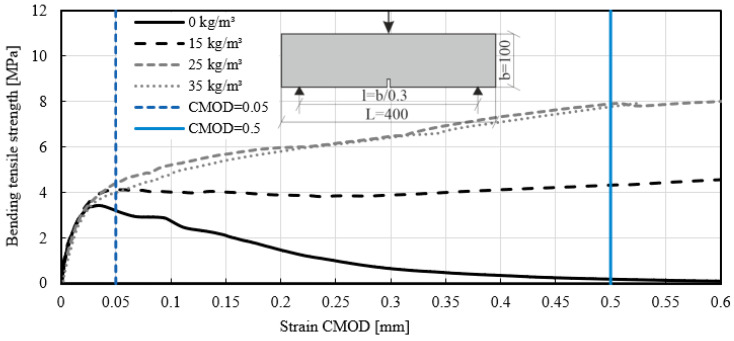
Stress–strain curves of the beam test-series with different fiber dosages MasterFiber 235 SPA (PP). Clip gauge strain in accordance with EN 14651 [39].

**Figure 13 materials-14-03736-f013:**
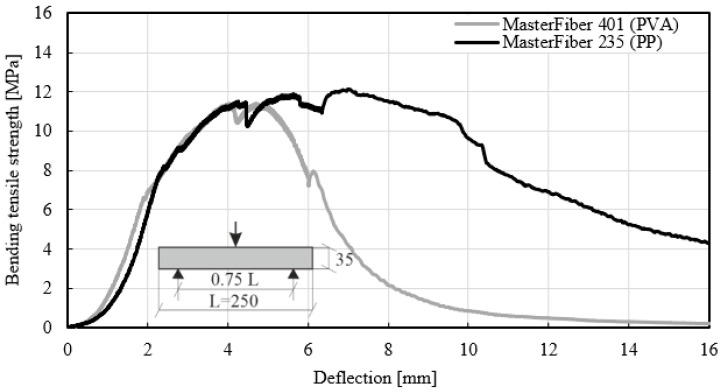
Bending-tensile strength of concrete with different fiber types (MasterFiber 235 SPA and MasterFiber 401) for a fiber dosage of 35 kg/m^3^ in accordance with EN 12467 [38].

**Figure 14 materials-14-03736-f014:**
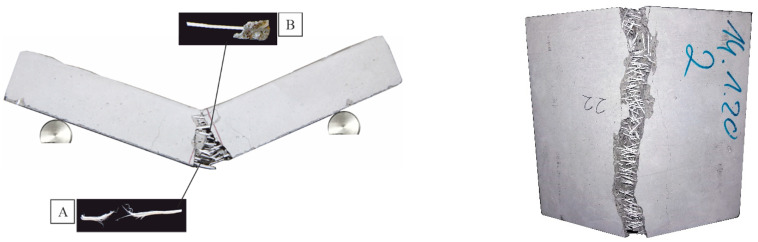
Typical failure modes for specimens [38] with MasterFiber 235 SPA (PP). (**A**) Images of the tensile fracture face of a fiber. (**B**) Images of the pull-out fracture face of a fiber.

**Figure 15 materials-14-03736-f015:**
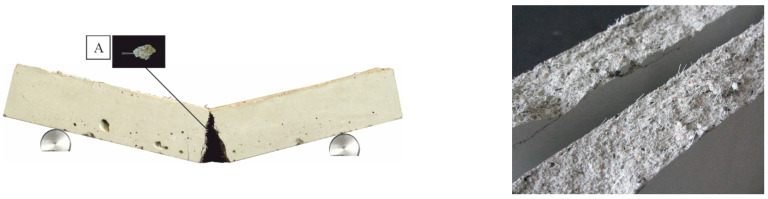
Typical failure modes for specimens [38] with MasterFiber 401 (PVA). (**A**) Images of the pull-out fracture face of a fiber.

**Figure 16 materials-14-03736-f016:**
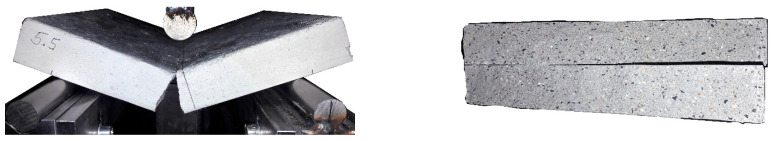
Typical failure modes for specimens [38] without fiber.

**Table 1 materials-14-03736-t001:** Mechanical parameters of the used fiber.

Type of Fiber	Tension Strength [MPa]	E-Modul [MPa]	Diameter [mm]	Length [mm]	Specific Gravity [kg/m^3^]
MasterFiber 235 SPA (PP)	500	>8.000	0.70	30	910
MasterFiber 401 (PVA)	800	29.000	0.16	12	1.300

**Table 2 materials-14-03736-t002:** Concrete mix design.

Material	Raw Density [kg/m^3^]	Weight [kg/m^3^]
Cement (CEM I 42.5 R)	3100	650
Aggregate 0 to 3 mm	2600	990
Silica fume	700	50
Limestone powder	2700	415
Plasticizer MasterGlenium ACE 430	1060	18
Water	1000	210

**Table 3 materials-14-03736-t003:** Overview of tested samples.

Series	Fiber Dosage [kg/m^3^]	Number of Specimens
panel specimens (250 × 250 × 35 mm^3^) in accordance with EN 12467 [38]	0	3
	15	3
	25	3
	35	3
beam specimens (100 × 100 × 400 mm^3^) in accordance with EN 14651 [39]	0	3
	15	3
	25	3
	35	3

**Table 4 materials-14-03736-t004:** Fresh concrete properties of the investigated concrete mixtures.

Mix ID	Type of Fiber	Fiber-Dosage	Air Void	Bulk Density [st. dev]	Slump Flow Diameter [st. dev]
		[kg/m^3^]	[%]	[kg/m^3^]	[mm]
1	MasterFiber 401 (PVA)	35	3.8	2288 [6]	418 [25]
2	MasterFiber 235 SPA (PP)	35	4.2	2239 [8]	650 [42]
3		25	3.8	2251 [21]	635 [21]
4		15	3.5	2278 [16]	595 [7]
5		0	2.8	2307 [20]	565 [7]

**Table 5 materials-14-03736-t005:** Bending tensile strength of concrete with different fiber dosages of MasterFiber 235 SPA (PP) in accordance with EN 12467 [38].

Mix ID	Type of Fiber	Fiber-Dosage [kg/m^3^]	$f_{L}$ [st. dev] [MPa]	Percentage Increase [%]
2	MasterFiber 235 SPA (PP)	35	6.62 [0.20]	118
3		25	6.43 [0.16]	115
4		15	5.85 [0.38]	104
5		0	5.61 [0.40]	100

**Table 6 materials-14-03736-t006:** Flexural tensile force according EN 14651 [39].

Mix ID	Fiber-Dosage	Type of Fiber	$f_{L}$ LOP	Percentage Increase [%]	$f_{L}$ at Prescribed CMODj Values in MPa, [st. dev]			
	[kg/m^3^]		[MPa]		0.5 [mm]	1.5 [mm]	2.5 [mm]	3.5 [mm]
2	35	MasterFiber 235 SPA (PP)	3.89 [0.07]	124	5.66 [0.14]	8.57 [0.07]	8.03 [0.10]	7.34 [0.11]
3	25		4.10 [0.06]	131	6.30 [0.08]	8.04 [0.09]	9.16 [0.04]	9.24 [0.01]
4	15		4.04 [0.03]	129	3.06 [0.15]	4.17 [0.14]	4.78 [0.14]	4.94 [0.10]
5	0		3.14 [0.09]	100	0.11 [0.22]	-	-	-

## Data Availability

Not applicable.

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
