# Peer review of "Influence of Fiber Addition on the Properties of High-Performance Concrete"

_materials, 2021, doi:10.3390/ma14133736_

Round 1
Reviewer 1 Report
The originality and the scientific value of the subject research can be better.
The research area is the Influence of Fiber Addition on the Properties of High-Performance Concrete.
The main part of the research is an experimental program.
The research area is focused on the mechanical properties of using high-performance concrete.
The experimental program is nicely done, but some information and conclusions about high-performance concrete are known.
However, the experimental program must have a greater informative value.
The manuscript must be edited and improved.
It was also appropriate to research the microstructure or structural elements of concrete, for example.
Extensive research is being solved in the given area of the application of high-performance concrete with case studies and is a lot of information from the solved area. The introduction section needs to be also improved.
Mateckova, P., et. al. Comparative Study of High-Performance Concrete Characteristics and Loading Test of Pretensioned Experimental Beams. Crystals 2021, 11, 427.
Abbas, S. et. al. Mechanical Behavior of Ultrahigh-Performance Concrete Tunnel Lining Segments. Materials 2021, 14, 2378.
Table 2 - please check
Concrete mix design - Specify values ​​for m3 of the concrete mix!
Describe in more detail the individual components of the concrete mix.
Figure 2 must be enlarged.
Part of the discussion must be mentioned separately.
Also, state standard deviations or VoC for all tests.
This applies to the entire manuscript
Figure 4. Add standard deviations or VoC to the figure.
This applies to the entire manuscript
Gave more attention to why the modulus of elasticity decreased. Please Explain.
Figure 8. Please enlarge the figure.
Also, add an image with a test record - Load-displacement diagram.
It would be appropriate to put LD diagrams / Bending ten./deflection diagram in the appendices of the article so that the results of the tests can be used by other scientists. Also, support the citation of the article.
Add a photo from the test in Figure 8.
Also, add an image with a test record - load-displacement diagram for the test in Figure 9.
It would be appropriate to put LD diagrams / Bending ten./deflection diagram in the appendices of the article so that the results of the tests can be used by other scientists. Also, support the citation of the article.
Figure 10. Improve the image caption. The strain is only for the range 0 to 0.35
You should have a dot in the image instead of a comma for decimal places.
Advanced 3D numerical modelling could be better used for detailed evaluation and analysis of experiments.
The manuscript has good structure, but part of the discussion is missing.
I recommend editing of manuscript:
1) The benefit for further research must be clearly defined in the article.
2) Rewrite the article with more criticism (discussion) of the results.
In summary, the manuscript can be significantly improved for the reader.
The document must be revised.
Reviewer 2 Report
The article analyzes the problem of using polymeric fibers within concrete, major revisions are needed for the article.
The title should be changed to mortar and not concrete as the coarse aggregate is not used.
Why have all the tests not been carried out also on the 15 and 25 percentages of the PVA fibers? The chimcal structure of PVA fibers is more compatible with the cement and should give better results, even if smaller in size.
In the explanation of the compression tests, reference is made to the formation of voids within the sample resulting from the increase in the quantity of fibers used, analyzing the table I noticed that the percentage of fibers in PVA results to have a lower void content compared to higher percentages of PP. I would therefore add in the text that the formation of voids is also linked to the little interaction between the PP and the cement paste which therefore nullifies the mechanical result.
This explanation with the formation of voids, however, is in contrast to what was stated in the lines 238-240, the PP is not wettable and has little interaction with the cement paste, in general we see a bridging effect mainly linked to the size of the fibers and not to the interaction with the cement paste. Otherwise it would be necessary to demonstrate at least with microscope images of the fracture area or with a pull-out test.
Table 6 does not show the C-MOD value of the pure mortar, why was this comparison ignored?
Using the CMOD it is also possible to calculate the fracture energy and therefore the energy dissipated by the sample during the break, why was it not calculated?
Reviewer 3 Report
This study intends to determine the effect of fiber dosage and type on the mechanical characteristics of HPFRC. The results are interesting. However, more explanations need to be added to the text to show the research gaps in this field. The introduction needs to be improved. Please consider the following comments to improve the structure of the manuscript:
- Abstract, Lines 11-17: Please summarize the general sentences in the abstract and concentrate more on experimental tests and results. Please add more quantitative results at the end of the abstract.
- Page 2, Lines 66-71: After carefully reading the introduction, the reviewer could not find the research gaps in this field. Please determine the main objectives of the present work in the last paragraph of the introduction based on the research gaps.
- Page 2, Lines 78-81: there are general sentences and should be transferred to the introduction. Section 2.1.1 should only explain the fibers used in this experimental study.
- Figure 1: Please add a scale bar.
- Table 2 should be modified to mention concrete compositions instead of raw densities of materials. Also, please mention w/c or w/b rations of mixtures.
- Page 5, Lines 160-162: these are conflicting explanations. Please modify.
- Page 6, Line 193: It should be Fig. 5 for compressive strength results. Please check the figure numbers throughout the manuscript.
- Page 6, Lines 184-185: it may be due to the higher air content in the fiber-reinforced mixtures compared to normal concrete. Adding fibers during mixing and pouring mixture within the molds should be checked to prevent fiber congestion in some areas, reducing the quality of concrete. Moreover, the dosage of fibers can affect compressive strength. The authors can explain why they selected the present dosages.
- Fig. 12: Please show failure modes for PVA, PP, and normal concrete to compare them.
- Page 11, Lines 328-329: this is not the results of the present study. Please remove it.
- Page 11, Line 323: Please remove “chapter” throughout the manuscript.
Round 2
Reviewer 1 Report
The research area and results are from the context of the manuscript can better understand.
Thanks for the comments and manuscript edits.
The manuscript has sufficient informational value.
The presentation of the research and results is also at a good level.
The manuscript can be accepted for publication.
Reviewer 2 Report
The authors answered all my questions, the article is acceptable for publication.
Reviewer 3 Report
The authors appropriately improved the structure of the manuscript based on the comments of the reviewer.